# Food Patterns of Hospitalized Patients with Heart Failure and Their Relationship with Demographic, Economic and Clinical Factors in Sergipe, Brazil

**DOI:** 10.3390/nu14050987

**Published:** 2022-02-25

**Authors:** Jamille Oliveira Costa, Juliana Santos Barbosa, Luciana Vieira Sousa Alves, Rebeca Rocha de Almeida, Victor Batista Oliveira, Larissa Monteiro Costa Pereira, Larissa Marina Santana Mendonça de Oliveira, Raysa Manuelle Santos Rocha, Diva Aliete dos Santos Vieira, Kiriaque Barra Ferreira Barbosa, Ingrid Maria Novais Barros de Carvalho Costa, Felipe J. Aidar, Márcia Ferreira Cândido de Souza, Joselina Luzia Menezes Oliveira, Leonardo Baumworcel, Eduardo Borba Neves, Alfonso López Díaz-de-Durana, Marcos Antonio Almeida-Santos, Antônio Carlos Sobral Sousa

**Affiliations:** 1Postgraduate Program in Health Sciences, Federal University of Sergipe (UFS), Aracaju 49060-676, Brazil; barbosa.juliana@live.com (J.S.B.); lucianaalvesnutri@gmail.com (L.V.S.A.); rebeca_nut@hotmail.com (R.R.d.A.); vbo.nutri@gmail.com (V.B.O.); larissa_monteiroo@hotmail.com (L.M.C.P.); nutrilarissamarina@gmail.com (L.M.S.M.d.O.); ysamanu@hotmail.com (R.M.S.R.); joselinamenezes@gmail.com (J.L.M.O.); acssousa@terra.com.br (A.C.S.S.); 2Department of Nutrition, Campus Prof. Antônio Garcia Filho, Federal University of Sergipe (UFS), Lagarto 49400-000, Brazil; divaaliete@academico.ufs.br; 3Department of Nutrition, Federal University of Sergipe (UFS), Sao Cristovao 49100-000, Brazil; kiribarra@yahoo.com.br; 4Food Technology Department, São Cristóvão Campus, Federal Institute of Sergipe, Sao Cristovao 49100-000, Brazil; ingrid_novais@infonet.com.br; 5Group of Studies and Research in Performance, Sport, Health and Paralympic Sports—GEPEPS, Federal University of Sergipe (UFS), Sao Cristovao 49100-000, Brazil; fjaidar@gmail.com; 6Postgraduate Program in Physical Education, Federal University of Sergipe (UFS), Sao Cristovao 49100-000, Brazil; 7Postgraduate Program Professional in Management and Technological Innovation in Health, Federal University of Sergipe (UFS), Aracaju 49100-000, Brazil; nutrimarciacandido@gmail.com; 8Department of Medicine, Federal University of Sergipe (UFS), Sao Cristovao 49100-000, Brazil; 9Division of Cardiology, University Hospital of Federal University of Sergipe (UFS), Sao Cristovao 49100-000, Brazil; 10Clinic and Hospital São Lucas/Division, Rede D’Or São Luiz, Aracaju 49060-676, Brazil; leonardo.baumworcel@caxiasdor.com.br (L.B.); marcosalmeida2010@yahoo.com.br (M.A.A.-S.); 11Postgraduate Program in Biomedical Engineering, Federal Technological University of Paraná (UTFPR), Curitiba 80230-901, Brazil; eduardoneves@utfpr.edu.br; 12Sports Department, Physical Activity and Sports Faculty-INEF, Universidad Politécnica de Madrid, 28040 Madrid, Spain; alfonso.lopez@upm.es; 13Postgraduate Program in Health and Environment, Tiradentes University (UNIT), Aracaju 49010-390, Brazil

**Keywords:** dietary patterns, cardiovascular diseases, cardiac insufficiency, food consumption, quality of life

## Abstract

Background: The high rates of hospitalization and mortality caused by Heart Failure (HF) have attracted the attention of health sectors around the world. Dietary patterns that involve food combinations and preparations with synergistic or antagonistic effects of different dietary components can influence the worsening and negative outcomes of this disease. Objectives: To describe the dietary patterns of patients hospitalized for HF decompensation and associate them with demographic, economic, and clinical factors, and the type of care provided in Sergipe. Materials and Methods: Cross-sectional study that is part of the Congestive Heart Failure Registry (VICTIM-CHF)” of Aracaju/SE. Prospective data collection took place with all patients hospitalized between April 2018 and February 2021 in cardiology referral hospitals, 2 public and 1 private. The data collected were sociodemographic, clinical, lifestyle, anthropometric and food consumption variables. Daily dietary intake was estimated by applying a semiquantitative food frequency questionnaire. The extraction of dietary patterns, by exploratory factor analysis, was performed after grouping the foods according to the nutritional value and form of preparation into 34 groups. To assess the association between the factorial scores for adherence to the standards and the variables studied, the Mann-Whitney U test was applied. Linear regressions were also performed, considering the dietary pattern (one for each pattern) as a dependent variable. Results: The study included 240 patients hospitalized for HF decompensation, most of them elderly (mean age 61.12 ± 1.06 years), male (52.08%) and attended by the Unified Health System—SUS (67.5%). Three dietary patterns were identified, labeled “traditional” (typical foods of the Brazilian northeastern population added to ultra-processed foods), “Mediterranean” (foods recommended by the Mediterranean diet) and “dual” (healthy foods combined with fast and easy-to-prepare foods like snacks, bread, sweets and desserts). Adherence to the “traditional” pattern was greater among men (*p* < 0.031) and non-diabetics (*p* < 0.003). The “Mediterranean” was more consumed by the elderly (*p* < 0.001), with partners (*p* = 0.001) and a lower income (*p* < 0.001), assisted by the SUS (*p* < 0.001) and without hypertension (*p* = 0,04). The “dual” diet pattern had greater adherence by the elderly (*p* < 0.001), self-declared non-black (*p* = 0.012), with higher income (*p* < 0.001), assisted in the private sector (*p* < 0.001) and with less impaired functional capacity (*p* = 0.037). It was also observed that being female (*p* = 0.031) and being older reduced the average scores of performing the “traditional” pattern (*p* = 0.002). Regarding the type of service, being from the public service reduced the average scores for adhering to the “dual” pattern (*p* = 0.008). Conclusions: Three dietary patterns representative of the population were found, called traditional, Mediterranean and dual, which were associated with demographic, economic and clinical factors. Thus, these standards must be considered in the development of nutritional strategies and recommendations in order to increase adherence to diets that are more protective against cardiovascular diseases.

## 1. Introduction

Heart Failure (HF) has gained priority attention in health sectors around the world [1,2]. This clinical syndrome, resulting from the heart’s inability to pump blood and supply nutrients to meet tissue demand [3], has been responsible for 21% of hospital admissions and 10.8% of the causes of mortality from cardiovascular diseases (CVDs) in Brazil [4]. It is estimated that, in 2030, more than 8 million people will have HF in the world [5] and will be victims of their psychological and physical limitations—which compromises the performance of daily routine activities and quality of life [6,7,8,9].

According to the Brazilian HF Directive (2018), the treatment of this disease, to minimize unfavorable clinical outcomes, involves pharmacological and non-pharmacological treatments, among which nutrition stands out. But, like other guidelines, the protocol adopts the approach focused on the intake of macro and/or micronutrients with a recommendation of low intake of simple carbohydrate, saturated or trans fat and sodium, prioritizing sources of complex carbohydrate, mono and polyunsaturated fat [10].

The literature has long recommended that the study of the influence of food on health or disease be done through dietary patterns, and not isolated nutrients [11,12]. After all, nutrition involves foods, combinations and preparations that result from synergistic or antagonistic effects of different dietary components. Assessing food consumption through standards has been widely used in children and adolescents [13,14,15], adults and the elderly [16,17,18] and population with CVDs, with the focus on adherence to the Mediterranean diet and Dietary Approaches to Stop Hypertension—DASH [19,20,21].

However, there are no articles in the literature that assess national and regional dietary patterns in patients hospitalized for decompensation with HF and its associated factors. Describing the items that make up these patterns and their effects on the protection or prediction of injuries can be a fundamental approach to propose nutritional strategies and recommendations for CVDs, considering eating habits more realistically and promoting greater adherence to the healthier pattern. Thus, this study aimed of to describe the dietary patterns of patients hospitalized for HF decompensation and associate them with demographic, economic and clinical factors and the type of care provided in Sergipe.

## 2. Materials and Methods

### 2.1. Study Design

The present study is of the cross-sectional type and is part of the Congestive Heart Failure Registry (VICTIM-CHF)” of Aracaju/SE, with the collection carried out from April 2018 to February 2021, in 3 reference hospitals in cardiology, being 2 public (Sergipe Emergency Hospital—HUSE and Hospital Surgery) and 1 private (Saint Luke’s Hospital, Aracaju, Sergipe, Brazil).

### 2.2. Participants

Data collection took place with individuals over 18 years of age, of both genders, hospitalized for HF decompensation, admitted and with a diagnosis given by two examiners of the hospital’s cardiology team using the Boston Score and Framingham criteria [22], and who maintained the diagnosis at the hospital discharge. Patients with other debilitating chronic diseases (such as human immunodeficiency virus, cancer or chronic obstructive pulmonary disease), difficulty in oral feeding, psychiatric disorders, or neurocognitive conditions were excluded.

As this is a study resulting from a Registry, prospective data collection was carried out from all patients, with an interview being carried out complemented with data collection from medical records, as described in Figure 1. Data were collected using a questionnaire developed for the present study, which contained sociodemographic, clinical, anthropometric, lifestyle and food consumption variables.

### 2.3. Sociodemographic, Clinical and Lifestyle Data

Sociodemographic variables of identification (age and gender), race, education, family income, presence of associated pathologies (hypertension, diabetes mellitus, dyslipidemia, Peripheral Arterial Disease (CAD), Chronic Kidney Failure and Cancer), type and etiological factor of the HF and functional capacity quantified by the New York Heart Association (NYHA), which stratifies, by the degree of physical limitation, into classes I, II, III and IV (NYHA, 1994). To perform the statistical analysis, patients were divided into 2 groups, NYHA I–II and NYHA III–IV.

To assess clinical cardiac function, Doppler echocardiography (ECHO) was used, performed at the inpatient hospital. The analyzed variable was the left ventricular ejection fraction (LVEF), categorized as preserved (>50%) and altered (<49%).

Information on smoking (smokers or non-smokers) and alcoholism were analyzed in terms of consumption (yes or no). Regular practice and physical activity intensity were assessed using the International Physical Activity Questionnaire (IPAQ), short version, an instrument recommended by the World Health Organization (1998) and validated in Brazil by the Center for Studies of the Aptitude Laboratory Physics of São Caetano do Sul—CELAFISCS [23].

### 2.4. Anthropometric

After stabilizing the patient in the hospital, weight and height were measured by the previously trained nutrition team, in triplicate, to minimize errors. Height was measured using a stadiometer (Seca ^®^, Hamburg, Germany), with markings in millimeters. To measure body weight, an electronic digital scale was used, with a maximum capacity of 180 kg and approximation of 100 g (Seca ^®^, Hamburg, Germany). For bedridden patients, weight and height were estimated by Chumlea, Roche and Mukherjee [24], considering gender, race and age. Edemas and ascites were discounted as recommended by James [25].

The Body Mass Index (BMI) was calculated by the ratio between body weight (kg) and squared height (m) and classified based on the cutoff points proposed by the Ministry of Health [26,27].

### 2.5. Food Intake

Food intake was assessed by applying the semi-quantitative Food Frequency Questionnaire (FFQ) in person, with 81 foods adapted for the study population, using the instrument developed by Furlan-Viebig and Pastor-Valero [28] to assess the relationship between diet and chronic diseases. The frequency of questioned intake (daily, weekly, monthly, annual or never) was related to the last year. The quantification of energy and nutrient intake was performed using the Table of Nutritional Composition of Foods Consumed in Brazil (POF) [29] or by the Brazilian Table of Food Composition (TACO) when no food was found in the first one [30].

To minimize limitations, such as defining the quantity of food consumed, some methodological precautions were adopted—such as the use of visual resources (photo albums, utensils and standard measures). In addition, interviewers were trained (nutritionists and nutrition students), and a pilot test to resolve doubts and standardization of household measures was taken.

### 2.6. Ethical Questions

The study was approved by the Research Ethics Committee of the Federal University of Sergipe (Opinion: 2,670,347). In accordance with the principles of the Declaration of Helsinki and after a clear explanation of the study protocol, each participant signed the Free and Informed Consent Term—FICF, allowing their voluntary participation.

### 2.7. Statistical Analysis

To obtain the dietary patterns of the population, food was grouped, according to the nutritional value, similarity of consumption and form of preparation, into 34 groups, which were subjected to exploratory factor analysis [31]. Table 1 describes the foods and their respective groups. After grouping the foods, the Kaiser-Meyer-Olklin (KMO) test was performed, which confirmed the existence and weight of partial correlations, indicating the adequacy of the analysis [32,33]. When performing the scree plot analysis, three types of dietary patterns representative for the study population were determined, which were subjected to an orthogonal rotation (varimax) to improve the interpretability of the matrix of factor loadings of the data. The factor loading greater than or equal to |0.30| determined the foods that significantly compose the dietary pattern [34]. To assess the association between the adherence factor scores for the patterns and the sociodemographic, clinical, lifestyle and nutritional characteristics, the Mann-Whitney U test was applied. Linear regressions were also performed to assess factors associated with adherence to retained dietary patterns, considering the dietary pattern (one for each pattern) as a dependent variable and the other variables as independent (age, BMI, sex, race, marital status, income, type of care, NYHA, physical activity, self-reported presence of arterial hypertension, diabetes mellitus, alcohol consumption and smoking). Accepted statistical significance was established at *p* < 0.05. All analyzes were conducted using the R statistical program.

## 3. Results

Table 1 describes the foods and their respective groups.

The study included 240 patients hospitalized for HF decompensation, most of them elderly (mean age 61.12 SD 1.06 years) and male. There was a predominance of self-declaration of the black race, individuals who live with their partners, with a family income of less than 1 minimum wage, admitted to hospitals of the Unified Health System. Health—SUS. Regarding the clinical characteristics, the patients had altered functional capacity, were hypertensive, overweight and engaged in light to moderate physical activity. Alcohol intake and smoking were not very prevalent in the population. Table 2 shows the sociodemographic, clinical and lifestyle characterization of the evaluated population.

Three dietary patterns were identified, according to the exploratory factor analysis (Table 3), labeled as “Traditional”, “Mediterranean” and “Dual”. The Traditional pattern consisted of foods such as pasta, rice, corn couscous, roots and tubers, potato, flour, cookies and cakes, vegetables, fruit juices, red meat, eggs, sausages, beans, oilseeds, sauces, sweets and desserts, snacks, sandwiches and açaí (positive charges). The Mediterranean pattern included bread, vegetables, fruits, olive oil (positive charge) and red meat, beans and soft drinks (negative charge). Finally, the Dual Diet with bread, soups, milk and dairy products, olive oil, sweets and desserts, sandwiches, açaí, tapioca (positive charge) and corn couscous, industrialized juice, white meat (negative charge). Together these patterns explain 27.60% of the total variance of the data.

Table 4 shows the relationship between factor scores for dietary patterns and socioeconomic, clinical, and lifestyle characteristics. Adherence to the “Traditional” dietary pattern was greater for men and non-diabetics. The “Mediterranean” was more consumed by the elderly, with partners, with lower income, assisted by the Unified Health System and without hypertension. The “Dual” diet pattern had greater adherence by the elderly, self-reported non-black, with higher income, assisted in the private sector and with less impaired functional capacity.

The associated demographic, economic and clinical factors and the type of care for each dietary pattern are described in Table 5. It is observed that being female and older has a lower score in the Traditional dietary pattern score. Regarding the type of care, being from the public service had lower scores for the dual pattern.

Table 6 shows the association between nutrient intake and factor scores for dietary patterns. It was observed that patients with greater adherence to the traditional pattern (3rd tertile) had a higher intake of fiber, saturated, mono, poly, trans fat, total sugar, added sugar and sodium when compared to the group with lower adherence (1st tertile). This was also observed for the Mediterranean and dual standards, except for sodium whose lower adherence had higher consumption in the dual standard.

## 4. Discussion

The eating habits of patients with HF in Sergipe were represented by three dietary patterns characterized by healthy and unhealthy foods and that were associated with demographic, economic and clinical variables.

The population studied showed characteristics similar to those found in surveys conducted in Brazil, Latin America and the USA [35,36,37]. The prevalence of elderly people hospitalized due to HF decompensation ranged from 70 to 73% in these records. The I Brazilian Registry of HF (BREATHE), carried out by Albuquerque et al. [35], included 1263 patients hospitalized in 51 centers in different regions, with a mean age of 64 ± 16 years, treated by the Unified Health System (64.8%), hypertensive (70.8%) and diabetic (34%). There was a discrepancy with the present study concerning race and gender, with the national registry showing a higher prevalence for women (60%) and self-reported white ethnicity (59%). However, the proportion of women found in Sergipe was similar to that of Paraíba (48%), Brazil (49%) [4] and the US trials (40% to 50%) [38,39].

The “Traditional” pattern is characterized by typical foods of the Brazilian northeastern population, with the presence of foods such as beans, rice, pasta, corn couscous, roots and tubers, eggs and cassava flour added to ultra-processed foods such as sweets, desserts, snacks and embedded. According to the Family Budget Survey (POF 2017–2018) [40], the Northeast has the second-highest frequency in the consumption of cassava flour in the country and the highest frequency of poultry food (37.4%), corn (25.8%), pasta and related preparations (23.4%), eggs (18.1%) and beans (13.5%). Greater adherence to this standard was responsible for the nutritional quality coming from all food groups, with the offer of healthy nutrients such as complex carbohydrates, fiber, vitamins, minerals and unsaturated fats, contrasted with simple sugars, saturated and trans fat, and sodium from snacks, desserts, sausages and meats, which are also part of the standard.

According to the Brazilian cardioprotective diet, developed by the Ministry of Health in partnership with the “Hospital do Coração”, the Western standard requires greater control of consumption from cardiac patients, as there is a higher frequency of foods belonging to the yellow and blue groups, which, despite being composed of fresh, minimally processed or processed foods, provide more energy, saturated fat, salt and cholesterol (blue), which can make treatment difficult [41]. Furthermore, according to the III Brazilian Guidelines on Chronic HF, the treatment of this clinical syndrome consists of adopting a food plan that prioritizes sources of complex carbohydrate, monounsaturated and polyunsaturated fat, in addition to restricting sodium-rich foods [3].

The lower adherence of women to the traditional pattern may be associated with the lack of care and nutritional knowledge by the male population [42]. Women are more concerned about food quality and seek to further analyze information from food labels such as ingredient lists, portion size, and product quality appeal [43]. In addition, the fact that preparation is historically attributed to women reduces the consumption of ready-to-eat meals and meals away from home [40].

National surveys show that men have a worse quality of food when compared to women. It is observed that they eat more ultra-processed foods, salt, soft drinks, milk and meat with excess fat and less in natura, or minimally processed, such as fruits and vegetables [40,44,45]. The study by Assumpção et al. [46], showed a difference between the Revised Diet Quality Index (IQD-R) for men and women, and men had higher scores in the components of fruits, vegetables and milk; while for women, the score was higher only for the meat and eggs components. These data corroborate Canuto et al. [47], who, in a systematic review of social inequalities in food consumption in Brazil, observed, in summary, that men ingest more typical foods of the Brazilian diet, added to foods considered to be at risk for non-communicable chronic diseases (NCDs). Women, on the other hand, have a more prudent food consumption, consisting of fruits, vegetables and natural juices.

In relation to age, the elderly tend to be more careful in their food intake due to concern with health and/or also because of the greater probability of presenting morbidities that need, in their treatment, adjustments in eating habits [48,49]. In addition, older people have food monotony that does not include, in most cases, processed and ultra-processed foods in their habits, as when they were formed there was low availability and were considered less digestive and inappropriate for their age, resulting in low consumption of these products [50]. When evaluating the quality of the diet in a population of public servants, Pires et al. [49] observed that people over 65 years of age and with low education had better scores on the highest quality scores. A similar result was verified by Beck et al. [51] when evaluating the dietary pattern and socioeconomic characteristics of the population of New Zealand. Elderly people show greater adherence to the “Healthy” pattern, consisting of breakfast cereals, skim milk, soy and rice milk, soup and broth, yogurt, bananas, apples, other fruits and tea, and low consumption of pies and sweets, French fries, white bread, take-out food, soft drinks, beer and wine.

The “Mediterranean” pattern is composed of foods that offer healthier nutrients such as complex carbohydrates, vitamins and minerals, polyunsaturated fat and proteins, and avoid the consumption of simple sugar from soft drinks and saturated fat from red meat. This pattern received this name because it is similar to the Mediterranean diet, whose recommended foods are fruits, vegetables, vegetables and olive oil and low intake of simple sugar from sweetened beverages [52]. As for the cardioprotective diet, it has more foods from the green group, classified as cardioprotective rich in vitamins, minerals, fiber and antioxidants [41].

Adherence to the Mediterranean diet has been associated not only with a decrease in the incidence but also in the outcomes of HF [19]. Tekonidis et al. [53] observed a 45% reduction in mortality from this disease for the highest quartile, compared to the lowest adherence score for this pattern. Fitó et al. [54] found that the consumption of this pattern, with the addition of virgin olive oil or walnuts, decreased the cerebral natriuretic peptide-BNP, an important marker of the prognosis of patients with HF, (*p* < 0.05) and oxidized LDL (mainly in the olive oil group, *p* = 0.004) compared to the control group.

Just like the “traditional”, the “dual” pattern, consisting of fast and easy-to-prepare foods such as snacks, bread, sweets and desserts, rich in simple carbohydrates, but also complex sugars and good fats from soup and olive oil, requires greater care during the treatment of cardiovascular events.

The substitution of saturated fat food sources for carbohydrate sources can be seen, which may be associated with the fact that nutritional recommendations and/or guidelines indicate a reduction in fat consumption. However, the type and amount of carbohydrates have a significant influence on the treatment of these pathologies, since saturated fat can be synthesized through lipogenesis again from acetyl CoA, a product of carbohydrate metabolism [55]. Lima et al. [54] observed that, in public health services and, especially in the private sector, the main nutritional guidelines for post-infarction patients were in relation to added salt, salty foods, red meat, sausages, fats and fried foods.

The fact that the lower-income population reduces the scores for adherence to the dual pattern, which is considered unhealthy, is already discussed in the literature. Studies show that having greater purchasing power is not necessarily linked to healthy eating [40,47]. Just as they have the opportunity to consume fresh foods with greater monetary value, they also consume a lot of processed foods and ready-to-eat meals. The latest family budget survey’s result (2017–2018) observed that the population with lower income classes consumed healthy foods such as rice and beans, sweet potatoes, cassava flour and corn in greater quantities when compared to individuals from higher-income classes [40]. This result corroborates with the present study, which found an association between the “Mixed” pattern with low income and the “traditional” pattern, consisting of healthy and unhealthy foods, with high income.

The study by Torreglosa et al. [56], when evaluating the direct cost of food with the quality of the diet of adults with CVDs in Brazil, found that there are no significant differences between food classified as of good and poorer nutritional quality, which can minimize barriers of adoption of healthy eating habits often taxed as expensive.

According to what was discussed and in order to minimize the consequences caused by HF, there is a need to implement strategies, which maintain good eating habits of women and the elderly, but also encourage the adoption of healthy standards by men, individuals without pathologies, without partners, with high income and who attended in the private health service.

The limitations of the present study that must be considered are common to cross-sectional studies that assess the population’s food consumption. They are: not allowing the establishment of causal relationships, but only associations between variables; loss of data from some variables or incomplete information; the bias of over or underestimation by instruments available in the literature that require the interviewee’s memory and that have inter- and intra-individual variability; the subjectivity of exploratory factor analysis, as the researcher decides on the grouping of foods, number of factors and the values of the significant factor loading. However, all decisions were based on methodological aspects published in scientific studies that discuss the nutritional epidemiology associated with dietary patterns. It is noteworthy that this study is a pioneer, nationally and regionally, in the assessment of food consumption a posteriori, with the determination of dietary patterns in patients with HF that must be analyzed and considered during the treatment of the disease. In addition, the findings of the present study show that the population made up of women and the elderly show better adherence to healthy eating patterns and that men, individuals without pathologies, without partners, with high income and assisted in the private health service need care to improve food and thus minimize the aggravations of the disease.

## 5. Conclusions

The population with decompensated HF in Sergipe is predominantly made up of men, elderly, black people, with a family income of less than 1 minimum wage and who are assisted by the SUS. They are patients with altered functional capacity, hypertensive, overweight and light to moderate physical activity practitioners. Alcohol intake and smoking were not very prevalent in the population.

Three dietary patterns representative of the studied population were found, denominated as traditional (marked by typical foods of the Northeastern diet with the offer of healthy and unhealthy nutrients), Mediterranean (composed of healthy foods indicated by the Mediterranean diet) and dual (quick and easy-to-prepare foods with unhealthy nutrient supply). These patterns were associated with demographic, economic and clinical factors. Being female and being of legal age reduced the scores for adherence to the traditional dietary pattern. Already being seen in the public service reduced the scores in the dual pattern score. By knowing the eating habits of the population studied, we can see the need to implement food and nutrition education strategies for cardiovascular diseases that encourage the adoption of healthier eating patterns, such as the Mediterranean, marked with foods that are more protective against diseases and limitations in quality. of life that CI can cause.

## Figures and Tables

**Figure 1 nutrients-14-00987-f001:**
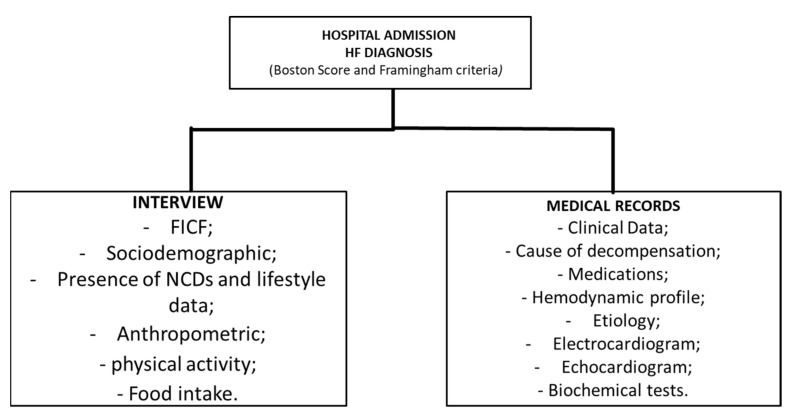
Data collection flowchart. HF—Heart failure; FICF—free and informed consent form; NCDs—non-communicable chronic diseases.

**Table 1 nutrients-14-00987-t001:** Grouping of foods according to nutritional value for analysis of dietary patterns.

Groups	Foods
Pasta	Pasta and pasta with sauce
Instant past	noodles and lasagna
Rice	White and whole
Breads	French (Jaco); slice; wholemeal; Brazilian cheese bread; potato bread
Corn couscous	corn couscous
Roots and tubers	Sweet potatoes; cassava and yam
English potato	Boiled English potatoes or mashed potatoes
Flour	Cassava and crumbs
Cookies and cakes	Water cracker; cornstarch cookie; cream cracker and basic cake
Green Vegetables	Lettuce; cabbage and kale
Vegetables	Tomato; beet; carrot; cucumber
Corn	Corn cob and canned
Soup	Soup
Juices	Pineapple; orange; tangerine; guava; watermelon; mango; grape; passion fruit; lemon; acerola; soursop; mangaba and umbu
Fruit	Pineapple; banana; orange or tangerine; guava; apple; papaya; watermelon; mango; grape and avocado
Industrialized juice	instant juice; energy drink and box juice
Red meat	Beef; pork; lamb; offal
White meat	Chicken and fish;
Eggs and frittata	Boiled or fried eggs; aratu; chicken or beef frittata
Processed meat products	Sausage; ham; salami and hot dogs
Milk derivatives	Whole; skimmed or semi-skimmed UHT milk; Whole; skimmed or semi-skimmed milk powder; whole or low-fat yogurt and ice cream Cheeses: rennet; cheddar; mozzarella; dish and butter curd.
Beans	Baked beans and feijoada
Legumes	Peas; lentils and chickpeas
Oilseed	Peanuts and Brazil nuts and cashew
Butter/margarine	Butter or margarine
Olive oil	Olive oil
Soft drinks	Regular; diet or light
Sauces	Ketchup; mayo and mustard
Sweets and dessert	Chocolate; brigadeiro; chocolate; candy; pudding; mousse; dulce de leche; guava; stuffed cake; stuffed biscuit and popsicle
Snacks	Potato chips; corn chips; party snacks; cereal bar; sweet and savory popcorn
Sandwiches	Hamburger; X-burger and grilled ham and cheese
Açaí	Açaí
Tapioca	Sweet and salty tapiocas
Coffee	Coffee with or without sugar or sweetener

**Table 2 nutrients-14-00987-t002:** Characterization of socio-demographic; clinical and lifestyle of patients hospitalized for decompensated HF in Aracaju, SE; 2021.

Variables	*n*	(%)
Age (*n* = 240)		
Adult (≤59 years)	108	45.00
Elderly (≥60 years)	132	55.00
Gender (*n* = 240)		
Male	125	52.08
Female	115	47.92
Race (*n* = 232)		
Not black	62	26.72
Black	170	73.28
Marital Status (*n* = 235)		
Single	109	46.38
With partner	126	53.62
Income (*n* = 240)		
<1 MW	164	68.33
≥1 MW	76	31.67
Type of HF by EF (*n* = 212)		
Changed	123	58.02
Preserved	89	41.98
Type of service (*n* = 240)		
Private	78	32.50
Public	162	67.50
NYHA (*n* = 240)		
1 and 2	30	12.50
3 and 4	209	87.50
Hypertension (*n* = 238)		
No	71	29.83
Yes	167	70.17
Diabetes Mellitus (*n* = 238)		
No	145	60.92
Yes	93	39.08
BMI (*n* = 196)		
No overweight	95	48.47
Overweight	101	51.53
Physical activity (*n* = 234)		
Absent/Mild/Moderate	198	84.62
High level	36	15.38
Alcoholic beverage (*n* = 238)		
No	203	85.2
Yes	35	14.7
Smoking (*n* = 238)		
No	227	95.38
Yes	11	4.62

MW—minimum wage; HF—Heart failure; EF—ejection fraction; NYHA—New York Heart Association; BMI—Body Mass Index.

**Table 3 nutrients-14-00987-t003:** Factor load of dietary patterns of patients hospitalized for HF decompensation in Sergipe (*n* = 240), 2021.

Food Groups	Traditional	Mediterranean	Dual
Pasta	0.42	−0.29	0.00
Instant noodles	0.10	0.16	0.12
Rice	0.40	−0.14	−0.18
Breads	0.28	0.38	0.30
Corn couscous	0.55	0.10	−0.35
Roots and tubers	0.32	0.12	−0.08
English potato	0.40	0.19	−0.32
Flours	0.41	−0.20	−0.14
Cookies and cakes	0.34	0.10	0.29
Green vegetables	0.50	0.53	−0.17
Vegetables	0.43	0.45	−0.05
Corn	0.20	0.06	−0.07
Soup	0.28	0.02	0.34
Juices	0.30	0.19	0.06
Fruits	0.37	0.30	−0.01
Industrialized juice	0.43	0.08	−0.39
Red meat	0.51	−0.30	−0.16
White meat	0.33	0.18	−0.43
Eggs and frittata	0.63	0.12	−0.08
Processed meat products	0.46	−0.08	0.00
Milk derivatives	0.27	0.02	0.84
Beans	0.56	−0.32	−0.11
Legumes	0.22	0.27	−0.11
Oilseed	0.35	0.08	0.11
Butter/margarine	0.20	−0.09	0.22
Olive oil	0.06	0.57	0.41
Soft drinks	0.37	−0.52	0.00
Sauces	0.35	0.06	−0.11
Sweets and dessert	0.64	−0.22	0.44
Snacks	0.63	−0.29	0.18
Sandwiches	0.37	−0.00	0.30
Açaí	0.32	−0.28	0.36
Tapioca	0.10	0.24	0.38
Coffee	0.22	−0.28	0.01
Explained variance (%)	11.24	10.01	6.35
Accumulated explained variance (%)	11.24	21.24	27.60
Kaiser-Meyer-Olkin (KMO)	0.72	-	-

Values are presented in factor loadings determined by exploratory factor analysis. The factor loading greater than or equal to |0.30| determined the foods that significantly compose the dietary pattern.

**Table 4 nutrients-14-00987-t004:** Factor scores of dietary patterns according to sociodemographic, clinical, lifestyle and nutritional characteristics of patients hospitalized with HF in Aracaju, SE, 2021.

Variables	Traditional			Mediterranean			Dual		
	Median	IIQ	*p*	Median	IIQ	*p*	Median	IIQ	*p*
Age (*n* = 239)			0.111			<0.001			<0.001
Adult (*n* = 108)	−0.14	−0.61; 0.60		−0.59	−0.47; 0.73		−0.35	−0.75; 0.17	
Elderly (*n* = 131)	−0.03	−0.59; 0.14		−0.40	−0.78; 0.15		0.18	−0.41; 0.68	
Gender (*n* = 239)			0.031			0.631			0.852
Male (*n* = 124)	−0.12	−0.58; 0.54		−0.16	−0.64; 0.39		−0.06	−0.62; 0.66	
Female (*n* = 115)	−0.32	−0.61; 0.14		−0.35	−0.68; 0.23		−0.04	−0.44; 0.33	
Race (*n* = 231)			0.955			0.131			0.012
Not black (*n* = 61)	−0.27	−0.80; 0.27		−0.40	−0.78; 0.31		0.17	−0.40; 0.61	
Black ( *n* = 170)	−0.24	−0.61; 0.32		−0.15	−0.63; 0.36		−0.16	−0.59; 0.33	
Marital Status (*n* = 234)			0.133			0.001			0.912
Single (*n* = 108)	−0.32	−0.62; 0.16		−0.40	−0.76; 0.12		−0.04	−0.58; 0.45	
With partner (*n* = 126)	−0.24	−0.52; 0.44		−0.01	−0.52; 0.69		−0.10	−0.52; 0.47	
Income (*n* = 239)			0.250			<0.001			<0.001
<1 MW (*n* = 164)	−0.26	−0.64; 0.25		−0.08	−0.55; 0.59		−0.25	−0.64; 0.24	
≥1 MW (*n* = 75)	−0.23	−0.58; 0.44		−0.47	−0.88; −0.06		0.33	−0.25; 0.76	
Type of HF by EF (*n* = 211)			0.938			0.059			0.711
Changed (*n* = 123)	−0.24	−0.60; 0.33		−0.19	−0.67; 0.45		−0.10	−0.58; 0.46	
Preserved (*n* = 88)	−0.27	−0.60; 0.45		−0.40	−0.80; 0.20		−0.04	−0.54; 0.60	
Type of service (*n* = 239)			0.811			<0.001			<0.001
Private (*n* = 77)	−0.23	−0.60; 0.26		−0.53	−0.83; −0.10		0.39	−0.00; 0.76	
Public (*n* = 162)	−0.26	−0.60; 0.33		−0.07	−0.51; 0.52		−0.34	−0.74; 0.22	
NYHA (*n* = 239)			0.358			0.627			0.037
1 and 2 (*n* = 30)	−0.11	−0.57; 0.44		−0.40	−0.69; 1.65		0.22	−0.31; 0.60	
3 and 4 (*n* = 209)	−0.26	−0.60; 0.32		−0.20	−0.67; 0.35		−0.11	−0.56; 0.44	
Hypertension (*n* = 237)			0.063			0.044			0.529
No (*n* = 71)	−0.13	−0.53; 0.56		−0.06	−0.59; 0.69		−0.19	−0.56; 0.34	
Yes (*n* = 166)	−0.29	−0.60; 0.19		−0.31	−0.74; 0.22		−0.04	−0.54; 0.47	
Diabetes Mellitus (*n* = 237)			0.003			0.494			0.808
No (*n* = 144)	−0.11	−0.53; 0.53		−0.16	−0.71; 0.56		−0.04	−0.48; 0.44	
Yes (*n* = 93)	−0.35	−0.71; −0.07		−0.35	−0.64; 0.15		−0.10	−0.58; 0.61	
BMI (*n* = 196)			0.909			0.775			0.557
No overweight (*n* = 95)	−0.26	−0.59; 0.32		−0.22	−0.77; 0.31		−0.11	−0.67; 0.36	
Overweight (*n* = 101)	−0.26	−0.60; 0.39		−0.31	−0.63; 0.37		−0.04	−0.46; 0.47	
Physical activity (*n* = 233)			0.516			0.372			0.327
Absent/Mild/Moderate (*n* = 197)	−0.26	−0.59; 0.23		−0.24	−0.67; 0.25		−0.09	−0.56; 0.39	
High level (*n* = 36)	−0.28	−0.59; 0.96		−0.11	−0.68; 0.76		0.09	−0.48; 0.78	
Alcoholic beverage (*n* = 237)			0.092			0.051			0.123
No (*n* = 202)	−0.27	−0.58; 0.23		−0.24	−0.69; 0.25		−0.03	−0.49; 0.47	
Yes (*n* = 35)	0.14	−0.62; 1.24		0.83	−0.46; 1.10		−0.36	−1.02; 0.36	
Smoking (*n* = 237)			0.300			0.732			0.199
No (*n* = 226)	−0.26	−0.60; 0.32		−0.23	−0.67; 0.31		−0.04	−0.52; 0.49	
Yes (*n* = 11)	0.14	−0.49; 0.72		−0.05	−0.87; 1.47		−0.20	−1.05; 0.33	

Values presented by factor scores. IIQ—Interquartile range; MW—Minimum wage; HF—Heart Failure; EF—Ejection Fraction; NYHA—New York Heart Association; BMI—Body Mass Index; *p* = significance level; teste de U de Mann-Whitney; significance level *p* < 0.05.

**Table 5 nutrients-14-00987-t005:** Demographic, economic and clinical factors and the type of care associated with dietary patterns of patients hospitalized with CHF in Aracaju, SE, 2021.

Variables	Traditional			Mediterranean			Dual		
	Β *	95%CI	*p*	Β *	95%CI	*p*	Β *	95%CI	*p*
Age	−0.02	−0.02; −0.01	0.002	−0.00	−0.02; 0.00	0.073	0.00	−0.00; 0.01	0.587
BMI	0.01	−0.01; 0.03	0.295	0.01	−0.01; 0.03	0.386	−0.14	−0.44; 0.15	0.345
Gender									
Male	Ref.			Ref.			Ref.		
Female	−0.33	−0.62; −0.05	0.031	0.11	−0.20; 0.43	0.475	−0.14	−0.44; 0.15	0.345
Race									
Not black	Ref.			Ref.			Ref.		
Black	0.00	−0.31; 0.32	0.953	0.03	−0.32; 0.39	0.847	−0.18	−0.52; 0.16	0.297
Marital status									
Single	Ref.			Ref.			Ref.		
With partner	−0.09	−0.37; 0.18	0.498	0.26	−0.03; 0.57	0.087	0.02	−0.27; 0.31	0.890
Income									
<1 MW	Ref.			Ref.			Ref.		
≥1 MW	0.17	−0.19; 0.54	0.356	−0.40	−0.82; 0.00	0.056	0.03	−0.36; 0.43	0.875
Type of service									
Private	Ref.			Ref.			Ref.		
Public	0.06	−0.33; 0.46	0.746	0.20	−0.23; 0.64	0.369	−0.57	−0.99; −0.15	0.008
NYHA									
1 and 2	Ref.			Ref.			Ref.		
3 and 4	0.10	−0.32; 0.54	0.620	−0.08	−0.57; 0.39	0.714	0.03	−0.42; 0.49	0.873
Physical activity									
Absent/Mild/Moderate	Ref.			Ref.			Ref.		
High level	0.19	−0.19; 0.57	0.323	0.29	−0.13; 0.72	0.182	0.00	−0.40; 0.42	0.967
Hypertension									
No	Ref.			Ref.			Ref.		
Yes	−0.12	−0.45; 0.20	0.455	−0.26	−0.64; 0.10	0.155	0.07	−0.28; 0.42	0.688
Diabetes Mellitus									
No	Ref.			Ref.			Ref.		
Yes	−0.25	−0.56; 0.04	0.092	0.17	−0.15; 0.51	0.295	−0.12	−0.44; 0.19	0.453
Alcoholic beverage									
No	Ref.			Ref.			Ref.		
Yes	0.08	−0.32; 0.49	0.684	0.25	−0.19; 0.71	0.267	−0.04	−0.48; 0.39	0.841
Smoking									
No	Ref.			Ref.			Ref.		
Yes	−0.13	−0.75; 0.48	0.663	−0.01	−0.70; 0.67	0.966	−0.55	−1.22; 0.10	0.098

MW—minimum wage; NYHA—New York Heart Association; BMI—Body Mass Index; Linear regressions; Significance level *p* < 0.05. * Coefficient β.

**Table 6 nutrients-14-00987-t006:** Median nutrient intake according to the adherence tertiles to dietary patterns of patients hospitalized with CHF in Aracaju, SE, 2021.

	Fiber (g)		Saturated Fat (g)		Monounsatured Fat (g)		Polyunsatured Fat (g)		Total Trans Fat (g)		Total Sugar (g)		Added Sugar (g)		Total Sodium (mg)	
	Median	IQR	Median	IQR	Median	IQR	Median	IQR	Median	IQR	Median	IQR	Median	IQR	Median	IQR
Traditional																
T1	23.36	13.07	20.82	17.19	21.32	10.99	10.45	4.69	2.23	2.10	110.02	72.90	25.23	27.73	1379.33	919.81
T2	32.32	17.41	27.74	19.39	27.58	16.39	12.98	6.64	3.34	1.95	134.10	71.72	32.42	31.87	1315.05	717.44
T3	42.51	25.30	45.83	22.99	42.68	23.02	21.76	11.79	6.49	4.60	223.91	124.18	84.28	59.47	1825.25	1358.28
*p*	<0.001 ^a,b,c^		<0.001 ^a,b,c^		<0.001 ^b,c^		<0.001 ^a,b,c^		<0.001 ^a,b,c^		<0.001 ^a,b,c^		<0.001 ^b,c^		<0.001 ^b,c^	
Mediterrnean																
T1	21.85	11.54	26.56	17.79	24.53	16.00	11.57	6.63	3.00	2.22	116.80	70.29	30.51	39.92	968.74	446.13
T2	28.77	13.79	26.05	20.47	23.63	13.84	12.34	6.36	2.94	2.49	134.71	85.17	36.69	45.14	1455.02	662.21
T3	46.06	21.78	41.77	22.93	42.68	22.83	22.22	12.63	4.83	4.79	223.36	131.65	63.98	82.27	2166.78	1263.46
*p*	<0.001 ^a,b,c^		<0.001 ^b,c^		<0.001 ^b,c^		<0.001 ^b,c^		<0.001 ^b,c^		<0.001 ^b,c^		<0.001 ^a,b,c^		<0.001 ^b,c^	
Dual																
T1	29.28	18.97	25.65	21.15	22.49	17.33	12.91	11.80	3.34	2.85	123.62	96.86	41.72	53.92	1744.32	1164.14
T2	25.53	15.22	26.56	14.37	23.74	10.16	11.55	4.72	2.92	2.45	125.02	81.32	34.62	49.63	1232.96	674.78
T3	41.38	24.63	44.52	22.21	42.23	20.96	17.74	10.90	4.81	5.30	195.57	127.79	41.58	78.95	1499.65	960.23
*p*	<0.001 ^b,c^		<0.001 ^b,c^		<0.001 ^b,c^		<0.001 ^b,c^		<0.001 ^b,c^		<0.001 ^b,c^		0.350		<0.001 ^a,b^	

Variables were described as median of tertiles. IQR—interquartile range; Chi-squared; Significance level *p* < 0.05. ^a^ significant difference between tertiles T1 and T2. ^b^ significant difference between tertiles T1 and T3. ^c^ significant difference between tertiles T2 and T3.

## Data Availability

The data that support this study can be obtained from the address: www.ufs.br/Department of Physical Education (accessed on 12 December 2021).

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
