# Peer review of "Food Patterns of Hospitalized Patients with Heart Failure and Their Relationship with Demographic, Economic and Clinical Factors in Sergipe, Brazil"

_nutrients, 2022, doi:10.3390/nu14050987_

Round 1
Reviewer 1 Report
In this cross-sectional, descriptive study in 240 patients hospitalized for heart failure in Sergipe, Brazil, Costa et al. observed three distinctive dietary patterns (traditional, Mediterranean, and dual) as well as heterogenous associations of these patterns with demographic, economic, and clinical indicators and the type of care provided. For example, men and non-diabetics were found to consume more of the traditional diet pattern, while the heathy Mediterranean pattern was commonly consumed by the elderly, those with partners, a lower income, assisted by the SUS or without hypertension. The dual dietary pattern had greater adherence by the elderly, self-declared non-black, with higher income, assisted in the private sector and with less impaired functional capacity. Based on these findings, the authors concluded that these standards must be considered in the development of nutritional strategies and recommendations in order to increase adherence to diets that are more protective against cardiovascular diseases.
Comments:
Abstract: The information given in the background section is lengthy, it doesn’t, however, clearly state what is the gap in knowledge that is being addressed in this study. There are some acronyms that are not defined and hard to understand: e.g., HR (VICTIM-CHF)” or SUS. The following 2 sentences may not be needed in the abstract: “The accepted statistical significance was p < 0.05. All analyzes were performed using the R statistical software.”. Instead it would be informative if the actual results are presented in a way, supported by numbers and p-values, at least for some important significant results. Currently, it is unknown which results are significant or non-significant.
Introduction: Page 2, line 97, what do you mean by “this methodology”? I assumed it was dietary pattern, but it isn’t very clear from the current text. Some references (e.g., 9, 10, 12) are in non-English; would it be more reader friendly if they were in English as well?
Methods: The data collection flowchart (Fig 1) should stand alone, with a figure legend. Explain what is “FICF”? Clinical data is listed in both boxes – and hard to know which data were collected via questionnaire/interview versus via medical records. Anthropometric data is listed in the box interview – why? I wonder what it would look like if there was a participant recruitment flowchart. I understand that the authors recruited patients from a Registry for this particular study and excluded those with other chronic diseases (HIV, cancer or COPD), difficulty in oral feeding, psychiatric disorders, or neurocognitive conditions (Page 3, lines 121-124). How many cases were detected and excluded for the reasons listed above?
Results: First paragraph -- please do not repeat the percentages (%) in the text as they are already in Table 2. I am curious what was the criteria for defining “Adults” versus “Elderly” for the age groups (Table 2). Aren’t elderly people adults as well? Table 2 could benefit from inclusion of a bit more details, may define MW, HF type by EF, etc. For BMI, “no-overweight” doesn’t make a sense – is it normal? Also wondering whether there were no patients with obesity, as it only states “overweight”. For Tables 3-6, adding a detailed footnote, including a brief description of statistical analyses used to obtain the results, could help readers to better understand the content.
Table 6: I am a bit confused about what the tertiles represent. Does the “T1” across all three dietary patterns (first column) represent the lowest adherence score group, vice versa, the “T3” represents the highest adherence score group within each dietary pattern? For the Mediterranean dietary pattern, the tertile 3 (T3), which I assumed was the group with the highest adherence to the Mediterranean diet (i.e., people who eat healthier than those in tertiles 1 or 2), has the highest amount of saturated fat (41.77 g), total sugar (223.36 g), added sugar (63.98 g), etc. Is this correct? Could the authors elaborate on this?
Discussion: The section is somewhat lengthy and at times it reads like a lecture, dissociated from the actual observations in this study (for example: p16, lines 340-350). The authors might consider reducing the text amount not directly related to the findings of the present study and discuss their own findings more in association with relevant available evidence. The limitations listed are fairly general that of associated with cross-sectional study design. Was there any particular limitation specifically relevant to this study? While the cohort is fairly well balanced in terms of gender, it consists predominantly of blacks and recruitment appeared to be done only in Sergipe. Could the authors comment on the generalizability of the findings to a broader population? Furthermore, novelty of the findings are hard to pinpoint. At the end of the discussion section, the authors notes that “this study is a pioneer, nationally and regionally, in the assessment of food consumption a posteriori, with the determination of dietary patterns in patients with HF that must be analyzed and considered during the treatment of the disease”. A bit more elaboration might help to elevate the novelty of the findings.
Conclusions: This section reads just like a brief summary of the findings. The writing may benefit from describing the public health significance of the findings or perhaps how this new knowledge would help advance the field. On the other hand, the abstract has a conclusion sentence, which was not included in the main text.
Minor comments: Define acronyms at first use and stick to them, e.g., CNCDs (not defined), SUS, Registry name, CVD, etc.
Author Response
We appreciate the suggestions and made all the requested adjustments, Respectfully

Reviewer 2 Report
In this study, the authors investigated the clinical features and dietary patterns among patients with heart failure in a specific are of Brazil. They divided patients into 3 groups based on dietary patterns, and compared their clinical features. I agree with the authors that economic and demographic features are associated with dietary pattern; however, there are several critical concerns listed below.
- First of all, the definition and threshold of 3 groups (traditional, Mediterranean, and dual) looks unclear. If this is something arbitrary, then it is difficult to find any importance in results of this study.
- It is not a surprise that dietary pattern is associated with demographic, economic, and other clinical factors.
- What are the unique findings in patients with HF?
- Sample size is too small to account for variety of etiologies of HF.
- Too many abbreviations, such as HR, SUS, and SE. All of them make us confusing.
Author Response

(The authors gave the same response as above.)

Round 2
Reviewer 2 Report
Thank you for your clarifications. I have no additional comment.